# The Effects of Resistance Training on Blood Pressure in Preadolescents and Adolescents: A Systematic Review and Meta-Analysis

**DOI:** 10.3390/ijerph17217900

**Published:** 2020-10-28

**Authors:** Carles Miguel Guillem, Andrés Felipe Loaiza-Betancur, Tamara Rial Rebullido, Avery D. Faigenbaum, Iván Chulvi-Medrano

**Affiliations:** 1Department of Physical and Sports Education, Faculty of Physical Activity and Sport Sciences, University of Valencia, 46010 Valencia, Spain; carlesmg7@gmail.com; 2University Institute of Physical Education, University of Antioquia, Medellín 1226, Colombia; andres.loaiza@udea.edu.co; 3Tamara Rial Exercise & Women’s Health, Newtown, PA 18940, USA; rialtamara@gmail.com; 4Department of Health and Exercise Science, The College of New Jersey, Ewing, NJ 08628, USA; faigenba@tcnj.edu; 5UIRFIDE (Sport Performance and Physical Fitness Research Group), Department of Physical and Sports Education, Faculty of Physical Activity and Sports Sciences, University of Valencia, 46010 Valencia, Spain

**Keywords:** children, youths, neuromuscular training, cardiovascular health, overweight, obesity

## Abstract

The aim was to systematically review and meta-analyze the current evidence for the effects of resistance training (RT) on blood pressure (BP) as the main outcome and body mass index (BMI) in children and adolescents. Two authors systematically searched the PubMed, SPORTDiscus, Web of Science Core Collection and EMBASE electronic databases. Inclusion criteria were: (1) children and adolescents (aged 8 to 18 years); (2) intervention studies including RT and (3) outcome measures of BP and BMI. The selected studies were analyzed using the Cochrane Risk-of-Bias Tool. Eight articles met inclusion criteria totaling 571 participants. The mean age ranged from 9.3 to 15.9 years and the mean BMI of 29.34 (7.24) kg/m^2^). Meta-analysis indicated that RT reduced BMI significantly (mean difference (MD): −0.43 kg/m^2^ (95% CI: −0.82, −0.03), P = 0.03; I^2^ = 5%) and a non-significant decrease in systolic BP (SBP) (MD: −1.09 mmHg (95% CI: −3.24, 1.07), P = 0.32; I^2^ = 67%) and diastolic BP (DBP) (MD: −0.93 mmHg (95% CI: −2.05, 0.19), P = 0.10; I^2^ = 37%). Limited evidence suggests that RT has no adverse effects on BP and may positively affect BP in youths. More high-quality studies are needed to clarify the association between RT and BP in light of body composition changes throughout childhood and adolescence.

## 1. Introduction

The treatment for hypertension is usually pharmacological and has shown to be effective in 50% of adult patients [1]. However, in younger populations pharmacological treatment should be reserved for those who present with persistent elevated blood pressure (BP) despite lifestyle modification [2]. Therefore, it is reasonable to investigate non-pharmacologic treatments for youth and to emphasize preventative strategies including regular physical activity. Resistance training (RT) has been suggested as an effective non-pharmacological treatment for the prevention and management of high BP in adults [3,4], yet little is known about the effects of RT on BP in children and adolescents (6–18 years of age) [3]. 

Research evidence has found that cardiovascular disease has its roots in childhood, with some reports of endothelial damage occurring early in life [5]. The prevalence of diagnosed primary pediatric hypertension is increasing [6,7]. Primary pediatric hypertension is the cardiovascular condition whereby systolic or diastolic BP values are > 95th percentile for boys and girls up to 12 years of age and > 130/80 mmHg for youth older than 13 years of age [6]. Primary pediatric hypertension (as early as 7 years of age) has been associated with pathophysiological changes that tracks into later stages [6,8]. Moreover, the prevalence of obesity is increasing among youth and it has been identified as a risk factor for elevated BP [9,10]. Thus, the prevention and management of obesity early in life should be a primary consideration for reducing the prevalence of pediatric hypertension [9]. Of note, data from diverse populations indicate that childhood BP is associated with BP later in life [11]. Therefore, early treatment and management are needed since accelerated weight gain in youth may increase the risk of elevated BP later in life [12]. Juonala et al. reported that overweight or obesity early in life was predictive of many comorbidities and found that youth who were overweight or obese but who became nonobese as adults had a cardiovascular risk profile that was similar to those who were never obese [13]. Therefore, maintenance of normal body weight in children and adolescents may prevent the clustering of hypertension and other cardiovascular disease risk factors in adulthood [11]. Body mass index (BMI) is the most commonly used surrogate measure of adiposity and screening tool for cardio-metabolic risk [5]. 

Along with weight maintenance, physical activity can improve BP levels in adults independently of pharmacological treatment [14]. A clinical report demonstrated a decrease in BP values of −5/8 mmHg in hypertensive adults following aerobic training [15]. Traditionally, research and clinical efforts have focused on aerobic training as a means of BP management. Recently, RT has gained attention as an important modulator of BP. Regular participation in RT has been found to reduce BP by −4 mmHg and −5 mmHg in hypertensive adults who performed dynamic and isometric RT, respectively [15]. 

In addition to increasing muscular strength, muscular power, and local muscular endurance, RT in youth has shown to produce many health benefits including improvements in cardiovascular fitness, body composition, bone mineral density, blood lipid profiles, insulin sensitivity, injury resistance, and mental health [16,17,18,19,20,21,22,23,24]. By definition, resistance training is a specialized method of conditioning that involves the use of different modes of training with a wide range of resistive loads including body weight exercises and free weights (barbells and dumbbells) [16]. Although the potential health benefits of RT in youth have been widely studied, there is limited understanding about the effects of RT on BP in children and adolescents. Several systematic reviews and meta-analyses have examined the positive effects of RT on BP values in adults. However, no previous systematic review has quantitatively examined the association between RT on BP and BMI in youth. Given this research gap, a systematic review was conducted to examine the literature regarding the effects of youth RT on systolic and diastolic BP. In addition, a meta-analysis of selected studies was conducted to quantitatively evaluate the effects of RT on systolic and diastolic BP, and BMI, in children and adolescents. Given the potential health-related benefits of RT in adults, we hypothesized that RT would also produce beneficial effects on BP and BMI values in youth. 

## 2. Materials and Methods 

We followed the recommendations described in the Cochrane Handbook for Systematic Reviews of Interventions version 5.1.0 [25]. Also, the PRISMA statement was used to guide the reporting this Systematic Review (SR) [26] and the protocol for this study was registered in the PROSPERO data base (CRD42020187686).

### 2.1. Data Sources and Searches

Four electronic databases were searched: PubMed, SPORTDiscus, Web of Science Core Collection and EMBASE to February 2020. No restrictions were set to either publication period or language. The search strategy contained keywords, MeSH terms and Boolean connectors such as AND and OR as follows: [(hypertension OR blood pressure) AND (children OR preadolescents OR youth) AND (“resistance training” OR “weight training” OR strength training”)]. Additionally, included studies and SR on similar topics were reviewed the reference list to find other Randomized controlled trials (RCTS that met the selection criteria.

### 2.2. Eligibility and Study Selection

After examining the search results, two blinded authors independently assessed the eligibility of all studies retrieved from the databases based on eligibility criteria. Studies were included if they met the following criteria according to patient/problem, intervention, comparison/control or comparator, outcome and study design (PICOS) methodology [25,26]: (i) participants were youth (6–18 years); (ii) the type of study was RCT, (iii) at least one group had to perform RT and (iv) developing RCTs were excluded from this Systematic Review.

### 2.3. Data Extraction and Quality Assessment

Subject characteristics (i.e., first author’s last name; year of publication, age, sex, BMI and training status) and exercise dose were systematically and independently reviewed by two authors (Table 1). For missing data, the correspondence author was contacted by email, requesting information of interest.

### 2.4. Risk of Bias of Individual Studies

Two review authors worked independently to assessed risk of bias by using domains described in the Cochrane Handbook for Systematic Reviews of Interventions, version 5.1.0 [25]. This set of domains is based on evidence of associations between potential overestimation of effect and the level of risk of bias of the article that may be due to aspects of sequence generation, allocation concealment, blinding, incomplete outcome data, selective reporting. Each criterion was rated as low, high, or unclear risk of bias.

### 2.5. Data Synthesis and Satistical Analysis 

Information on the outcomes of interest was stored in a database. The main results for this study were SBP, DBP and BMI. For continuous outcomes, the group size, the mean values and the standard deviation (SDs) was recorded for each group compared in the included studies. Pooled effects were calculated using an inverse of variance model, and the data were pooled to generate a mean difference (MD) in millimeters of mercury (mmHg) and kilograms on meter squared (kg/m^2^) with corresponding 95% confidence intervals (95% CIs). All the studies for each outcome reported data in the same units, so it was possible to pool all studies regardless of whether they reported change data or final data. Significance was set at *p* < 0.05. Statistical heterogeneity was evaluated using the I^2^ statistic and classified according to the Cochrane Handbook [25]: negligible heterogeneity, 0% to 40%; moderate heterogeneity, 30% to 60%; substantial heterogeneity, 50% to 90%; and considerable heterogeneity, 75% to 100%. A random-effects model was used. All analyses were performed by one reviewer using Review Manager Version 5.4 and checked against the extracted data by one author. 

## 3. Results

### 3.1. Literature Search and Article Selection

Initial database searches yielded a total of 1269 articles and the remaining 21 RCTs were found in other sources. After performing screening by title and abstract, and then removing duplicates, a total of 405 research papers were discarded, thus obtaining a total of 32 RCTs for full-text review. Subsequently, 8 RCTs were included in the qualitative synthesis [27,28,29,30,31,32,33,34]. Finally, one author did not respond with missing data, therefore, that study was excluded from the quantitative synthesis. In total 7 studies were included in the meta-analysis [28,29,30,31,32,33,34] (Figure 1). 

### 3.2. Study Characteristics

Eight studies were included in the qualitative analysis, with a total of 8 intervention groups and 571 normotensive or pre-hypertensive youth (intervention group, n = 278; control group, n = 293). The mean of age was 13.28 (2.49) years. Of these, two studies were conducted only with normal weight subjects [27,34], one did not specific it [30], and five realized with obese subjects [28,29,31,32,33]. The mean of body mass index was 17.26 (35.7) kg/m^2^. In addition, in only two of the studies the subjects followed nutritional guidelines [28,31]. One study included only male subjects [29], and another reported the inclusion of exclusively females [27]. The remaining six studies included both sexes. Since blood pressure was not the primary outcome in most studies, there was a great heterogeneity in the measurement procedures. Two studies used a standard sphygmomanometer with cuffs [31,34], two others used an automatic model [28,30] where one was semi-automatic [27] and the other studies did not specify measurement device [29,32,33]. Moreover, significant heterogeneity in the protocols was found ranging from 6 [31] to 40 [30] weeks of RT (Table 1).

### 3.3. Risk of Bias Individual Studies

Three articles clearly report the method of random assignment to the groups [30,33,34]. Only two RCTs describe the allocation concealment [29,33]. In particular, three included studies reported blinding of outcome assessor, the remaining five were judged with unclear risk of bias [32,33,34]. Additionally, the 8 included RCTs do not describe blinding of study staff and study participants and were judged at high risk of bias for that domain [27,28,29,30,31]. Additional data from the individual analysis of risk of bias is presented in Figure 2.

### 3.4. Principle Findings

The results of the meta-analysis showed that no statistically significant reductions were found on the SBP [MD: −1.09 mm Hg (95% CI: −3.24, 1.07), P = 0.32; I^2^ = 67%] and the DBP [MD: −0.93 mm Hg (95% CI: −2.05, 0.19), P = 0.10; I^2^ = 37%] when comparing the RT groups to the control groups (P = 0.32; P = 0.10, respectively). However, compared to the control group, RT reduced BMI statistically significantly [MD: −0.43 kg/m^2^ (95% CI: −0.82, −0.03), P = 0.03; I^2^ = 5%]. Forest plots are presented in Figure 3, Figure 4 and Figure 5.

## 4. Discussion

The aim of this systematic review and meta-analysis was to quantify the effect of RT on the values of SBP, DBP and BMI in youth. To the best of our knowledge, this is the first systematic review with a subsequent meta-analysis that investigates the effects of RT on BP values in children and adolescents. While other studies have investigated the role of physical activity on cardiometabolic health in youth [35,36], no previous reports have examined the influence of RT in this population. As shown in previous research [3,37], RT has been found to offer observable health-related benefits in adults. Thus, we hypothesized that RT would have positive effects on BP and BMI in youth. 

Our main findings are that RT resulted in non significant reductions in SBP (−1.09 mmHg; P = 0.32) and DBP (−0.93 mmHg; P = 0.10) and statistically significant reductions in BMI (−0.43 kg/m^2^; P = 0.03) in youth. Although the research reports in this review failed to show statistical significance in terms of the ability of RT to lower systolic and diastolic BP, several factors need to be considered. These factors include the design of the RT protocols (i.e., training intensity, volume, frequency and duration) as well as the health status (all were normotensive) and the training status of the participants. Conflicting findings from several studies are likely due to differences in outcomes measures, study designs and study populations. Regarding the RCTs examined in our review, researchers used different RT protocols. While three studies performed RT with bodyweight exercises [28,29,30], one used sandbags and dumbbells [34], the others used weight machines [27,31,32,33]. Notable, there was wide variation in the prescription of RT variables including intensity, volume, frequency or duration. For instance, some protocols proposed two weekly sessions of high RT (12 RM) [34] while others trained 4 days per week with a moderate to high intensity (8 RM) [33]. Further, two studies added nutritional guidelines along with the RT protocol [28,31]. Interpretation and comparison of results would be more accurate with similar RT protocols and with subgroup analysis (i.e., obese and normal body weight; hypertensive and normotensive). There were also differences in the configuration of the control groups among studies that could have impacted the outcomes. For example, two studies did not advise participants about extra physical activity at school or in community based programs [29,33]. In one study that included adolescents who were obese, the control group consisted of adolescents who were not obese [27]. This aforementioned report showed moderate and substantial heterogeneity values in DBP and SBP, respectively (37% and 67%). The heterogeneous values found in this study could help explain why no statistically significant changes were found in SBP and DBP values following RT [27].

RT is an evidence-based preventative exercise intervention strategy that can promote health and well-being through the life course [37,38]. The benefits of progressive RT on muscular strength, muscular power, and local muscular endurance of children and adolescents is well described in several meta-analysis [39,40,41]. Moreover, RT has shown to produce many health-related benefits including improvements in cardiovascular fitness, body composition, bone mineral density, blood lipid profiles, insulin sensitivity, injury resistance and mental health improvements [16,18,19,20,21,22,23,24,42]. Longitudinal studies have confirmed the inverse relationship between low levels of strength early in life and risk of cardiovascular disease later in life [38,43,44,45]. Therefore, it seems plausible that RT could lower BP concurrent with improvements in other health markers. Some studies have speculated that the reduction in BP following RT in youth might be due to an increase in skeletal muscle mass which, in turn, may lead to a myocardial relaxation [32], diastolic filling peak velocity at the mitral septal annulus [32], an improvement in autonomic modulation [27] and/or an enhanced endothelial function [34]. In obese children, functional and structural cardiac abnormalities (i.e., increased left ventricle and left atrium dimensions, diastolic and systolic left ventricle, and right ventricle dysfunction) have been described in comparison to normo-weight children [46,47]. In this sense, BP mechanisms might be different. Further studies are needed in order to clarify the hypothetical link between RT and BP improvements in youth.

Our findings show a statistically significant improvement in BMI (P = 0.03) after an RT intervention. It has been established that exercise interventions can alter body composition (e.g., increase fat free mass) while BMI can remain the same or in some cases increase due to the increase in muscle mass [48]. Indeed, some studies demonstrated no change of BMI following RT despite the remarkable benefits on other health parameters such as endothelial function [49,50]. Therefore, BMI values may underestimate the effectiveness of RT interventions with respect to cardiovascular disease risk [51].

This study has several limitations that should be acknowledged; (1) the lack of systematic quantification of the RT intensity, volume or exercise selection; (2) BP was not a primary outcome in many of the studies included in the analysis; (3) heterogeneity in the outcome measurement procedures; (4) most of the RCTs analyzed did not adequately perform or report random sequence generation, allocation concealment and blinding of outcome assessment and; (5) moderate and substantial values of heterogeneity on SBP and DBP were found.

Although limited research has examined the effects of RT on BP in youth, our results suggest that RT does not have an adverse effect on the BP of children and adolescents and may be beneficial in lowering BP and improving BMI in this young population. Unfortunately, our findings do not allow for a recommendation on a specific dose of RT for effectively managing BP in youth. Nevertheless, a technique-driven and progressive RT program including multijoint exercises that involve the large muscle groups should be considered in the design of youth physical activity programs [16]. Further research is needed to effectively examine the “dose response” (e.g., intensity, volume, frequency) of youth RT interventions while exploring novel modes of RT like low intensity isometric handgrip exercise [52,53].

## 5. Conclusions

The present shows that there is limited data to determine the effects of RT on BP values in youth, while significant improvements in BMI have been demonstrated. Although the studies show a tendency towards reducing systolic and diastolic BP, the heterogeneity of the RT intensity, volume, frequency or duration make the interpretation of results difficult. Mechanisms by which RT may induce favorable adaptations in BP in youth are speculative. More high-quality studies are needed to clarify the association between RT and BP in youth with and without clinical conditions. 

## Figures and Tables

**Figure 1 ijerph-17-07900-f001:**
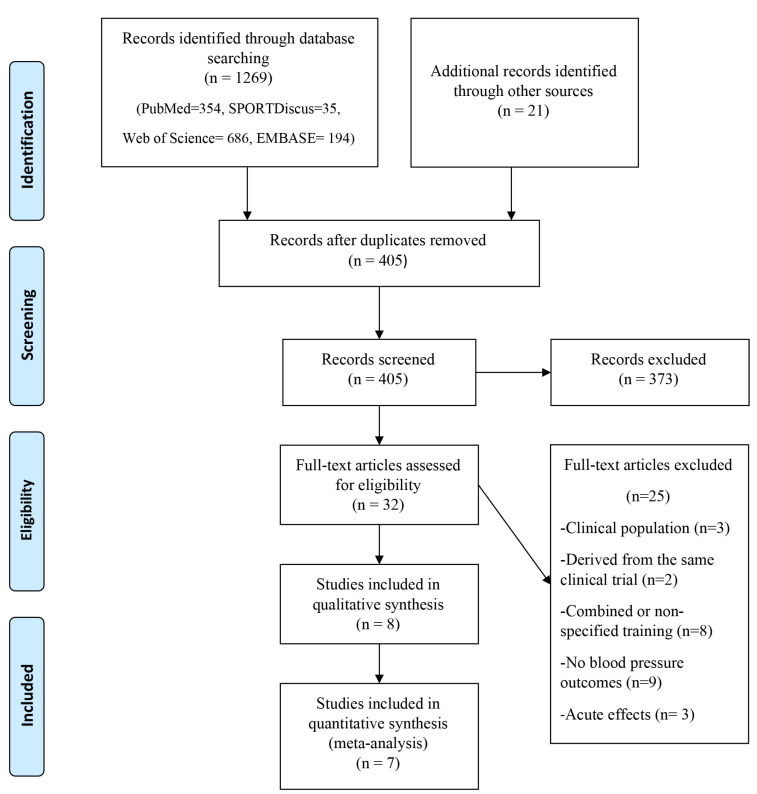
Preferred Reporting Items for Systematic Reviews and Meta-analysis (PRISMA) flow-chart of the study selection.

**Figure 2 ijerph-17-07900-f002:**
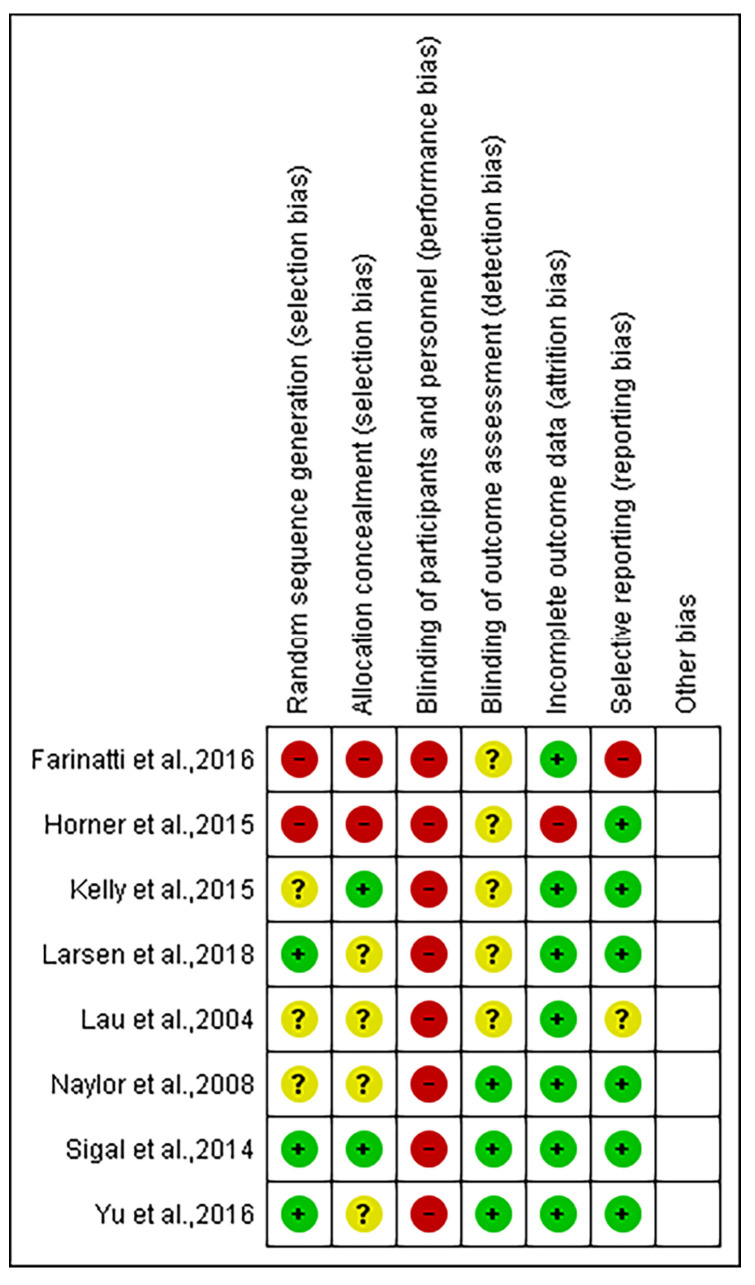
Summary of the risk of bias for the trials included in this meta-analysis. Green indicates low risk of bias, yellow indicated unclear, and red indicates high risk of bias.

**Figure 3 ijerph-17-07900-f003:**
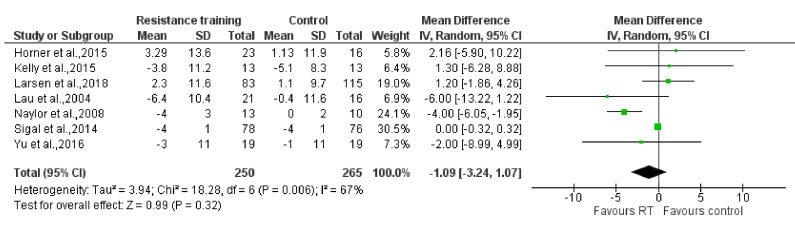
The effect of Resistance Training on systolic blood pressure (mmHg). Total: total number of subjects; CI: confidence interval.

**Figure 4 ijerph-17-07900-f004:**
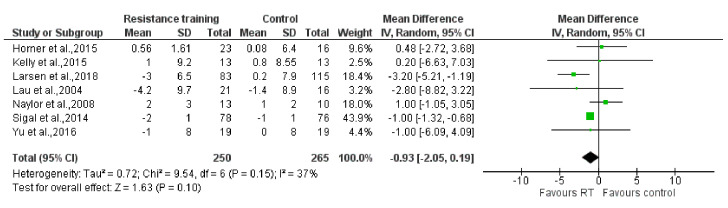
The effect of Resistance Training on diastolic blood pressure (mmHg). Total: total number of subjects; CI: confidence interval.

**Figure 5 ijerph-17-07900-f005:**
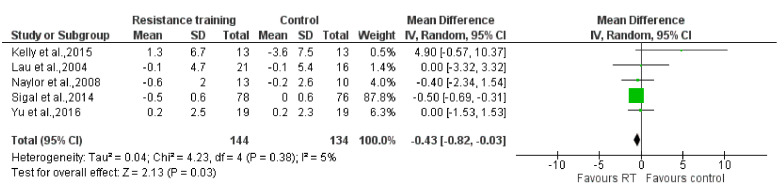
The effect of Resistance Training on body mass index (kg/m^2^). Total: total number of subjects; CI: confidence interval.

**Table 1 ijerph-17-07900-t001:** Resistance Training Studies with Blood Pressure outcome measures.

Source	Population	Intervention Description	BP Assessment Method	Frequency (D/WK)	Intensity	Volume (Sets × REPS)	Study Length (WKS)
Farinatti et al., 2016 [27]	Enrolled: N = 44 Completers: N = 44 44 F; Age: 13–17 Resistance group: N = 24. Obese	RT = chest and leg press, low row, leg extension, upper back, leg and arm curls, leg abduction/adduction, triceps ext.	Semi-automatic sphyngomanometer	3	1–2 Wks: 50–70% 10 RM 3–6 Wks: 60–80% 10 RM 7–12 Wks: 70–85% 10 RM	1 × 15 2 × 8–12 3 × 6–10	12
	Control group: N = 20. Non-obese						
Horner et al., 2015 [28]	Enrolled N = 81; Completers N = 66 41 M; 40 F; Age: 12–18 Resistance group: N = 27; 14 M 13 F; Age: 14.6 (1.9)	RT = Body exercises	automated sphygmomanometer	3	Not report	2 × 12	12
	Control group: 24 N = 24; 12 M 12 F; Age: 14.9 (1.8)						
Kelly et al., 2015 [29]	Enrolled N = 26; Completers N = 26 26 M; Age = 14–18 Obese Resistance group: N = 13; Age: 15.4 (0.9)	RT = day 1 consisted of compound lower body exercises and isolated upper body exercises and day 2 included com- pound upper body exercises and isolated lower body exercises.	Not report	2	1–4 Wks: light to moderate intensity 5–10 Wks: mod to high intensity) 11–16 Wks: mod to high intensity	1 × 10–15 2–3 × 13–15 3–4 × 8–12	16
	Control group: N = 13; Age: 15.6 (0.96)						
Larsen et al., 2018 [30]	Enrolled N = 83; Completers N = 83 Age = 8–10 Resistance group: N = 83	CST = Plyometric and dynamic strength exercises using upper and lower body.	automated sphygmomanometer	3	Not report	30-s all-out exercise periods with 45-s rest periods with 6–10 stations	40
	Control group: N = 115						
Lau et al., 2004 [31]	Enrolled N = 36; Completers N = 36 24 M; 12 F; Age = 10–17 Obese. Resistance group: N = 21	RT = Lat pull-down, shoulder press, leg press, leg extension, leg curl, heel raise, biceps curl, triceps extension, push-up.	standard mercury sphyngomanometer	3	75–85% RM	1 × 5	6
	Control group: N = 16						
Naylor et al., 2008 [32]	Enrolled N = 23; Completers N = 23 11 M; 12 F; Age = 12–14 Obese. Resistance group: N = 13; 7 M; 6 F Age: 12.2 (0.4)	RT = weight-stack machines.	Not report	3	75–90% RM	2 × 8	8
	Control group: N = 10; 4 M; 6 F; Age: 13.6 (0.4)						
Sigal et al., 2014 [33]	Enrolled N = 304; Completers N = 229 91 M; 213 F; Age = 14–18 Obese Resistance group: N = 78; 23 M; 55 F; Age: 15.9 (1.5)	RT = weight machines	Not report	4	65–85% RM	2 × 15	24
	Control group: N = 76; 24 M; 52 F; Age: 15.6 (1.3)						
Yu et al., 2016 [34]	Enrolled N = 38; Completers N = 38 25 M; 13 F; Age = 11–13 Non-obese. Resistance group: N = 19; Age: 12.3 (0.42)	RT = Elbow extension, elbow flexion, trunk extension, trunk flexion, shoulder press, knee extension, knee flexion, push-up, squats, incline dip and hip abd	standard sphygmomanometer	2	12 RM	3 × 12	10
	Control group: N = 19; Age: 12.1 (0.3)						

Abbreviations: N, simple size; Female (F), 290; Male (M), 218, RT, resistance training; CST, Circuit Strength Training; D, days; WK, week, WKS, Weeks; REPS, repetitions; RM, maximum repetitions

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
