# Peer review of "The Effects of Resistance Training on Blood Pressure in Preadolescents and Adolescents: A Systematic Review and Meta-Analysis"

_ijerph, 2020, doi:10.3390/ijerph17217900_

Round 1
Reviewer 1 Report
The work has been well written. Please find my minor comments below.
For general readers, it may be good to address a bit why RT has been suggested as an effective non-pharmacological treatment for the prevention and management of high BP in adults (L.37-39). How about endurance training (or aerobic training)? Can it also contribute to preventing or managing high BP similar to RT?
Author Response
Dear Editor, the authors appreciate the comments for our manuscript ijerph-963481. Our manuscript has been improved with the reviewer’s comments. Changes in our manuscript have been highlighted in red in order to facilitate the location of the suggested changes. We have addressed all reviewer’s concerns and included a point-by-point response below:
Reviewer 1:
For general readers, it may be good to address a bit why RT has been suggested as an effective non-pharmacological treatment for the prevention and management of high BP in adults (L.37-39). How about endurance training (or aerobic training)? Can it also contribute to preventing or managing high BP similar to RT?
Response:
Thank you very much. We added the following to the introduction: “Along with weight maintenance, physical activity can improve BP levels in adults independently of pharmacological treatment [15]. A clinical report demonstrated a decrease in BP values of -5/8 mmHg in hypertensive adults following aerobic training [16]. Traditionally, research and clinical efforts have focused on aerobic training as a means of BP management. Recently, RT has gained attention as an important modulator of BP. Regular participation in RT has been found to reduce BP by -4 mmHg and -5 mmHg in hypertensive adults who performed dynamic and isometric RT, respectively [16].
As for the question raised, the topic of the manuscript is resistance training thus we strongly believes that endurance/aerobic training should also be cited.
Reviewer 2 Report
As a reviewer I have the following remarks.
- Page 77: SR =?, spell first time abbreviation.
- Line 78: (#######). ?
- Figure 3-5. Your forest plot. I am guessing that the analysis was done in Stata. I think will be better to show large bars. Say reduce rang of x-axis to -15, 15.
- Figure 5. The effect of Resistance Training on body mass index (mmHg). Total: total number of subjects; CI: confidence interval. Sure “(mmHg)”?
- Thank you.
Author Response
Dear Editor, the authors appreciate the comments for our manuscript ijerph-963481. Our manuscript has been improved with the reviewer’s comments. Changes in our manuscript have been highlighted in red in order to facilitate the location of the suggested changes. We have addressed all reviewer’s concerns and included a point-by-point response below:
Reviewer 2:
Page 77: SR =?, spell first time abbreviation.
Response: Thank you for your comment. We have made the clarification according to the proposed suggestion.
Line 78: (#######)?
Response: Thank you for your question. Although registration of PROSPERO was requested, a definitive response has not yet been obtained, so we have decided to write down the temporary registration number. As soon as we have the registration it will be published.
CRD42020187686
Figure 3-5. Your forest plot. I am guessing that the analysis was done in Stata. I think will be better to show large bars. Say reduce rang of x-axis to -15, 15.
Response: Thank you for your consideration, unfortunately, we do not have neither license to nor the experience to use Stata. However, we take account the consideration for next meta-analysis. We have done the reduction of x-axis to -15, 15
Figure 5. The effect of Resistance Training on body mass index (mmHg). Total: total number of subjects; CI: confidence interval. Sure “(mmHg)”?
Response: Thank you for your comment, we have made the clarification according to the proposed suggestion.
Reviewer 3 Report
* COMMENT 1: Title and aim, in abstract and in introduction section, it´s not exactly the same. It´s necessary to shown the same during the paper, and according to prospero registry :
Title: “The Effects of Resistance Training on Blood Pressure in Youth: A Systematic Review and Meta-Analysis”.
The aim in the abstract: “The aim was to systematically review and meta-analyze the current evidence for the effects of resistance training (RT) on blood pressure (BP) and body mass index (BMI) in children and adolescents”.
Aim and hypothesis in introduction section ( pag 2, lines 66-73), “However, no previous
systematic review has quantitatively examined the association between RT on BP and BMI in youth.
Given this research gap, a systematic review was conducted to examine the literature regarding the
effects of youth RT on systolic and diastolic BP. In addition, a meta-analysis of selected studies was
conducted to quantitatively evaluate the effects of RT on systolic and diastolic BP in children and
adolescents. Given the potential health-related benefits of RT in adults, we hypothesized that RT
would also produce beneficial effects on BP values in youth.”
In prospero register (CRD42020187686) the Review question and the primary outcome are related to the blood pressure (BP), and the BMI is an “Additional outcome”. But the aim in the abstract don´t differentiate these and seems that both (BP and BMI) are “Main” outcomes:
The association between RT on BMI in youth is a gap to try to solve with your systematic review and meta-analysis but this is not reflected in the objectives and in the hypothesis at the of the introduction section. Please, unify or clarify these information in the paper.
*COMMENT 2: Keywords are incomplete, please complete: Overweigh, obesity or similar (Mesh term). Add a term related to BMI.
*COMMENT 3, pag 2, line 47: please, correct this sentence; “Moreover, the prevalence of obesity is increasing among youth has been identified as a risk factor for elevated BP”, Please change: “Moreover, the prevalence of obesity is increasing among youth and it has been identified as a risk factor for elevated BP”
*COMMENT 4, pag 2, line 77: …reporting this SR.
In pag 2, line 68 is the first time you include the term “systematic review” but the abbreviation in lacking. Please, add the abbreviation the first time.
*COMMENT 5, pag 3, line 109
The same with SBP and DBP. Please, add the abbreviation the first time.
*COMMENT 6, pag 2, line 80-81: “Four electronic databases were searched: PubMed, SPORTDiscus, Web of Science Core 80 Collection and EMBASE to February 2020”
- In prospero registry dates are different:
Date of registration in PROSPERO 05 July 2020,Date of first submission 26 May 2020.
Please, correct this information and perform again the search to be sure there is no publication that must be included between February and July 2020.
*COMMENT 7 pag 3, line 95-97 and Table 1:
-Subject characteristics (i.e. first author’s last name; year of publication, age, sex, BMI, training status and differences in the means of two time points on blood pressure with corresponding standard deviations)… (Table 1).
All these information is not included in table 1: differences in the means of two time points on blood pressure with …. Is lacking in table 1.
-Table 1: in population, it will be interesting to add information regarding Control group and intervention group. You specify M/F, enrolled and completers, age and obese/non obese, but I Think it´s important to indicate in each study how many participant were include in each group (intervention and control).
-Table 1: F290, M, 218: it´is necessary to include the meaning of the abbreviation F and M.
- Pag 4, line 136: ……, one did not specific it [29]”
But in table 1, reference 29, in population, you indicate “obese”. Please correct this discrepancy.
- Pag 4, line 139, ……reported the inclusion of exclusively females [26]”.
This information is lacking in Table 1, please add.
- Pag 4, line 142, ……where one was semi-automatic [26]”
But in Table 1: “ Not report”.
Please, correct.
- Table 1, ref 33, intervention description: …..push-up, squats, incline dip and hip abd”.
Please, add in abbreviation section the meaning of “ abd”.
*COMMENT 8: Please, improve the quality of the figures 3,4 and 5. It seems pixelated
*COMMENT 9 :Figure 4. The effect of Resistance Trainingon…”.
Please, correct “Trainingon”
*COMMENT 10 , lines 243, 244.. youth RT interventions while exploring novel modes of RT including low intensity isometric handgrip exercise [51,52].
- References 51 and 52 are examples of novel models RT, there are more options. Correct the sentence as follows or delete:
.. youth RT interventions while exploring novel modes of RT like low intensity isometric handgrip exercise [51,52].
*COMMENT 11 , conclusions section, lines 246-251:
Conclusion regarding the effects or RT on BMI are lacking. Please, complete conclusions section.
Author Response
Dear Editor, the authors appreciate the comments for our manuscript ijerph-963481. Our manuscript has been improved with the reviewer’s comments. Changes in our manuscript have been highlighted in red in order to facilitate the location of the suggested changes. We have addressed all reviewer’s concerns and included a point-by-point response below:
Reviewer 3:
COMMENT 1: Title and aim, in abstract and in introduction section, it´s not exactly the same. It´s necessary to show the same during the paper, and according to prospero registry:
Title: “The Effects of Resistance Training on Blood Pressure in Youth: A Systematic Review and Meta-Analysis”
Response: Change done
The aim in the abstract: “The aim was to systematically review and meta-analyze the current evidence for the effects of resistance training (RT) on blood pressure (BP) and body mass index (BMI) in children and adolescents”.
Response: Change done
Aim and hypothesis in introduction section ( pag 2, lines 66-73), “However, no previous systematic review has quantitatively examined the association between RT on BP and BMI in youth.
Response: Change done
Given this research gap, a systematic review was conducted to examine the literature regarding the effects of youth RT on systolic and diastolic BP. In addition, a meta-analysis of selected studies was conducted to quantitatively evaluate the effects of RT on systolic and diastolic BP in children and adolescents. Given the potential health-related benefits of RT in adults, we hypothesized that RT would also produce beneficial effects on BP values in youth.”
Response: Change done
In prospero register (CRD42020187686) the Review question and the primary outcome are related to the blood pressure (BP), and the BMI is an “Additional outcome”. But the aim in the abstract don´t differentiate these and seems that both (BP and BMI) are “Main” outcomes:
Response: Change done
The association between RT on BMI in youth is a gap to try to solve with your systematic review and meta-analysis but this is not reflected in the objectives and in the hypothesis at the of the introduction section. Please, unify or clarify these information in the paper.
Response: Change done
Thank you for your appreciations. We have checked the text and we have changed in order to be exactly the same text in the manuscript and in the PROSPERO register
COMMENT 2: Keywords are incomplete, please complete: Overweigh, obesity or similar (Mesh term). Add a term related to BMI.
Response: Thank you for your comment. We added the key words BMI-related as you suggested.
COMMENT 3, pag 2, line 47: please, correct this sentence; “Moreover, the prevalence of obesity is increasing among youth has been identified as a risk factor for elevated BP”, Please change: “Moreover, the prevalence of obesity is increasing among youth and it has been identified as a risk factor for elevated BP”
Response: Thank you for your comment. We have corrected the manuscript as suggested.
COMMENT 4, pag 2, line 77: …reporting this SR. In pag 2, line 68 is the first time you include the term “systematic review” but the abbreviation in lacking. Please, add the abbreviation the first time.
Response. Thank you for your comment. We have corrected the manuscript as suggested.
COMMENT 5, pag 3, line 109 The same with SBP and DBP. Please, add the abbreviation the first time.
Response. Thank you for your comment. We have corrected the manuscript as suggested.
COMMENT 6, pag 2, line 80-81: “Four electronic databases were searched: PubMed, SPORTDiscus, Web of Science Core 80 Collection and EMBASE to February 2020” - In prospero registry dates are different: Date of registration in PROSPERO 05 July 2020,Date of first submission 26 May 2020. Please, correct this information and perform again the search to be sure there is no publication that must be included between February and July 2020.
Response. Thank you for your comment. We have re-run the search as you suggest. We did not find any new inclusive research.
COMMENT 7 pag 3, line 95-97 and Table 1:
Subject characteristics (i.e. first author’s last name; year of publication, age, sex, BMI, training status and differences in the means of two time points on blood pressure with corresponding standard deviations)… (Table 1). All these information is not included in table 1: differences in the means of two time points on blood pressure with …. Is lacking in table 1.
Response. Thank you for your appreciation. We have made the changes according to your suggestion.
Table 1: in population, it will be interesting to add information regarding Control group and intervention group. You specify M/F, enrolled and completers, age and obese/non obese, but I Think it´s important to indicate in each study how many participant were include in each group (intervention and control).
Response. Individuals who were included in both the intervention and control groups have been included. However, it is important to note that in some studies there were more than 2 groups (e.g., aerobic), so the third group was not included.
We have include descriptive data in control groups of each included primary research- if there is not any data is due to that in the original published articles that data is missed.
Table 1: F290, M, 218: it´is necessary to include the meaning of the abbreviation F and M.
Response. Thank you for your appreciation. We have made the change according to your suggestion.
“Pag 4, line 136: ……, one did not specific it [29]” But in table 1, reference 29, in population, you indicate “obese”. Please correct this discrepancy.
Response: Thank you for your appreciation. We have made the change according to your suggestion.
Pag 4, line 139, ……reported the inclusion of exclusively females [26]”. This information is lacking in Table 1, please add
Response: Thank you for your appreciation. We have made the change according to your suggestion.
Pag 4, line 142, ……where one was semi-automatic [26]” But in Table 1: “ Not report”. Please, correct.
Response: Thank you for your appreciation. We have made the change according to your suggestion.
Table 1, ref 33, intervention description: …..push-up, squats, incline dip and hip abd”. Please, add in abbreviation section the meaning of “ abd”.
Response: Thank you for your appreciation. We have made the change according to your suggestion.
COMMENT 8: Please, improve the quality of the figures 3,4 and 5. It seems pixelated
Response: Thank you for your comment, we have made the tables according to the rules of the magazine (300dpi).
COMMENT 9: Figure 4. The effect of Resistance Trainingon…”. Please, correct “Trainingon”
Response: Thank you for your comment. We have made the change according to your suggestion.
COMMENT 10 , lines 243, 244.. youth RT interventions while exploring novel modes of RT including low intensity isometric handgrip exercise [51,52].
- References 51 and 52 are examples of novel models RT, there are more options. Correct the sentence as follows or delete
- ..youth RT interventions while exploring novel modes of RT like low intensity isometric handgrip exercise [51,52].
Response: Thank you for your comment. We have made the change according to your suggestion.
*COMMENT 11 , conclusions section, lines 246-251: Conclusion regarding the effects or RT on BMI are lacking. Please, complete conclusions section.
Response: Thank you for your comment. We have added the effects of RT and BMI in the conclusion as suggested.